# Gut Microbiota and Gastrointestinal Symptoms in the Global Assessment of Obsessive–Compulsive Disorder: A Narrative Review of Current Evidence and Practical Implications

**DOI:** 10.3390/brainsci14060539

**Published:** 2024-05-24

**Authors:** Giacomo Grassi, Ilenia Pampaloni

**Affiliations:** 1Brain Center Firenze, 50144 Florence, Italy; 2National OCD and BDD Unit, South West London and St Georges NHS Trust, London SW17 7DJ, UK; ilenia.pampaloni@swlstg.nhs.uk

**Keywords:** OCD, microbiome, microbiota, gut, irritable bowel syndrome, inflammatory bowel disease, assessment, probiotics, inflammation

## Abstract

A growing body of literature suggests a link between bowel syndromes (e.g., irritable bowel syndrome and inflammatory bowel disease), gut microbiome alterations, and psychiatric disorders. This narrative review aims to explore the potential role of the gut microbiome in the pathogenesis and clinical presentation of obsessive–compulsive disorder (OCD) and to explore whether there is sufficient evidence to warrant considering gastrointestinal symptoms and their implication for the gut microbiome during the assessment and treatment of OCD. For this purpose, a PubMed search of studies focusing on OCD, gut microbiota, irritable bowel syndrome, and inflammatory bowel disease was conducted by two independent reviewers. While the current literature on gut microbiome and gastrointestinal issues in OCD remains limited, emerging evidence suggests gut microbiome alterations and high rates of bowel syndromes in this population. These findings emphasize the importance of incorporating comprehensive gastrointestinal assessments into the “global assessment of OCD”. Such assessment should encompass various factors, including gastrointestinal physical comorbidities and symptoms, nutritional habits, bowel habits, fluid intake, exercise patterns, and potential microbiome dysfunctions and inflammation. Considering the treatment implications, interventions targeting gut health, such as probiotics and dietary modifications, may hold promise in improving symptoms in OCD patients with comorbid gastrointestinal problems. Further research in this area is warranted to better understand the interplay between gut health and OCD and to explore the effectiveness of targeted interventions in improving clinical outcomes.

## 1. Introduction

Obsessive–compulsive disorder (OCD) is a common and profoundly debilitating neuropsychiatric condition affecting approximately 1.3% of the general population [1]. OCD is characterized by obsessions and compulsions. Obsessions are intrusive, repetitive, and distressing thoughts, images, or urges that are often perceived as unrealistic or excessive by the individual experiencing them. Compulsions are repetitive and often ritualized behaviors or mental acts that individuals feel compelled to perform in response to an obsession or according to rigidly applied rules. The primary purpose of compulsions is to alleviate the anxiety or distress triggered by obsessions. These symptoms lead to significant distress or impairment, are time-consuming, and the individual often tries to resist or control them. Its pervasive impact extends beyond symptomatology, often leading to significant disability and a diminished quality of life [2,3]. The average duration of untreated illness has been estimated at 17 years [4]. If left untreated, OCD tends to have a chronic course [5]. OCD frequently presents with complex phenotypic manifestations, compounded by both physical and psychiatric comorbidities. The heterogeneous nature of OCD not only complicates diagnosis but also significantly impacts treatment response, underscoring the need for a comprehensive, holistic, and evidence-based approach to screening and assessment [6,7]. Such an approach must encompass a broad spectrum of considerations, including physical and psychiatric comorbidities (to include consideration about inflammation and autoimmunity) as well as psychological and functional assessment. Additionally, it should extend to lifestyle factors such as diet, nutrition, exercise, fluid intake, etc. Together, these elements constitute a “Global assessment of OCD”, as proposed by Pampaloni and colleagues [6].

It is known that the gut microbiome exerts extensive reciprocal interactions with the brain through microbial metabolites, the vagus nerve, and hormonal and immunological signaling, collectively forming the microbiome–gut–brain axis [8]. This emerging field holds promise for exploring new pathways through which the gut microbiota affects brain function, such as modulating neurotransmitter levels and influencing inflammatory and immune responses. These interactions may subsequently influence the development and maintenance of psychiatric conditions. Understanding these mechanisms could provide insights into developing alternative therapeutic strategies targeting the microbiome, including the use of probiotics, prebiotics, fecal transplants, and dietary interventions.

Epidemiological and clinical findings consistently support the association between irritable bowel syndrome (IBS) (a syndrome associated with gut microbiome perturbations) and psychiatric disorders, highlighting a significant prevalence of psychiatric comorbidities among individuals with IBS [9,10]. Moreover, recent systematic reviews and meta-analyses examining differences in the gut microbiome of patients with psychiatric disorders and healthy controls have revealed a decreased abundance of certain bacteria producing butyrate in psychiatric disorder cohorts compared to healthy individuals [11,12].

These observations raise intriguing questions about the potential influence of psychiatric symptomatology on gut microbial composition and vice versa. Although the studies cited above, along with several others, have demonstrated an involvement of the gut microbiome in various psychiatric disorders, its specific role in OCD remains not fully understood.

The authors chose to focus on the role of the gut microbiome in OCD for several key reasons. Understanding this relationship could be crucial for patients suffering from OCD, as a substantial proportion fail to respond to existing treatments. Investigating the role of the gut microbiome in OCD could lead to more effective and comprehensive treatment strategies, potentially benefiting those patients who have not responded to treatment or who continue to experience significant residual symptoms. Additionally, the symptomatology and phenomenology of OCD, including its comorbidities, provide strong grounds for such an investigation. OCD symptoms may contribute to the onset and maintenance of alterations in the gut microbiome through various mechanisms. For instance, compulsions such as excessive washing and cleaning can reduce exposure to diverse microbial species encountered in the environment on a daily basis [13]. Furthermore, OCD is often associated with altered eating habits, which can influence microbiome composition. Eating disorders show comorbidity rates with OCD of 17% [14], with a lifetime comorbidity of 19% in Anorexia Nervosa and 14% in Bulimia Nervosa [15]. Selective and restricted food intake (picky eating) has also been associated with OCD [16]. Conversely, individuals exhibiting picky eating behaviors that meet the diagnostic criteria for Avoidant/Restrictive Food Intake Disorder (ARFID) tend to exhibit a higher prevalence of OCD symptoms [17].

Additionally, patients with OCD may engage in elaborate and time-consuming compulsions following bowel movements, often resulting in intentionally decreasing the frequency of bowel movements. This reduction may be accompanied by food restriction or alterations in diet, serving as avoidance strategies. For instance, Chen et al. [13] demonstrated reduced fiber intake among patients with OCD. Similarly, patients with OCD may restrict fluid intake, which, besides increasing the risk of chronic kidney failure [18], can cause constipation and subsequently impact the composition of the gut microbiome.

This narrative review aims to explore the potential role of the gut microbiome in the pathogenesis and clinical presentation of OCD and to explore whether there is sufficient evidence to warrant considering gastrointestinal symptoms and their implication for gut microbiota during the assessment and treatment of OCD. This question stems from the recognition of OCD’s heterogeneous nature, its significant impact on both physical and psychological well-being, and the emerging understanding of the microbiome–gut–brain axis in psychiatric disorders. By exploring the relationship between OCD symptomatology and gut microbial composition, we aim to contribute to the development of more comprehensive, holistic, and personalized approaches to the assessment and management of OCD.

We hypothesize that OCD symptomatology, such as excessive cleaning and food and fluid restriction, may contribute to alterations in the gut microbiome and that specific changes in gut microbiota composition may, in turn, contribute to OCD. These changes could potentially serve as biomarkers for targeted interventions. This would make the assessment of eating habits and gastrointestinal symptoms an essential component of the “global assessment” of OCD. Additionally, we propose that interventions aimed at modulating the gut microbiome, such as dietary adjustments, probiotics, and prebiotics, may alleviate certain OCD symptoms and improve overall treatment outcomes. Further research in this area could pave the way for novel, microbiome-targeted therapies that complement existing OCD treatments.

## 2. Materials and Methods

In this narrative review, we aim to include all the available data in the field of OCD, gut microbiota, irritable bowel syndrome, and inflammatory bowel disease. For this purpose, a PubMed search was conducted by two independent reviewers (GG and IP). The search was carried out in December 2023, was updated in May 2024, and was not restricted regarding publication year. A broad search string was used covering multiple terms for OCD (“obsessive-compulsive disorder” OR “obsessions” OR “compulsions”) combined through the Boolean operator AND with terms related to microbiome, irritable bowel syndrome, and inflammatory bowel disease (“microbiome”, “gut”, “microbiota”, “gastrointestinal problems”, “irritable bowel syndrome”, “IBS”, “Inflammatory bowel disease”, “IBD”, “Crohn’s disease”, and “ulcerative colitis”). Inclusion criteria were original research articles on OCD and microbiome, OCD and irritable bowel syndrome, or OCD and inflammatory bowel disease. We included all articles on these topics, regardless of whether they were animal studies, genetic studies, or studies on clinical populations. Exclusion criteria were non-original articles (e.g., reviews) and articles that did not examine the link between OCD, microbiome, irritable bowel syndrome, and inflammatory bowel disease. Also, articles in a language other than English and published in non-peer-reviewed journals were excluded. The screening process, by applying those criteria to the title/abstracts of the articles, was carried out by the two authors (GG and IP). An agreement could be achieved concerning the few discrepancies in title/abstract screening. This resulted in 28 articles being selected and inserted in a shared Excel sheet for full-text screening, which reduced the final number of eligible articles to 19. It was further decided to group the articles by methods employed in these studies as they are content-wise extremely heterogeneous. This decision led to the following sections: clinical studies of gut microbiota, preclinical studies of gut microbiota, clinical studies on irritable bowel syndrome and OCD, and clinical studies on inflammatory bowel disease and OCD.

## 3. Results

### 3.1. Clinical Evidence of Gut Microbiota Alteration in OCD

In the last few years, a limited but consistent amount of literature has shown a gut microbiota alteration in OCD patients. In 2020, a case–control study on 21 unmedicated and non-depressed adult OCD patients was the first one describing a microbiota alteration in OCD subjects compared to healthy controls [19]. In that study, OCD patients showed a lower diversity (lower alpha diversity index) of gut microbiota species and a lower abundance of three butyrate-producing genera (Oscillospira, Odoribacter, and Anaerostipes) compared to a sample of matched healthy controls [19]. These results are intriguing since the reduced diversity of gut microbiota species has been previously linked to several other psychiatric disorders, such as depression, autism spectrum disorder, and PANS/PANDAS [20,21,22]. Also, the relatively lower abundance of butyrate-producing species has been related to inflammation and gut barrier integrity. Indeed, butyrate has been shown to exert anti-inflammatory effects and to be a relevant substrate for the colic epithelial cells, and therefore, it has a relevant role in the promotion of gut barrier integrity [23]. This latest factor is of high relevance when considering that the compromission of the gut barrier integrity is one of the leading hypotheses on the mechanistic link between gut dysbiosis and brain disorders.

A recent study consistently found a trend toward lower alpha diversity in the stool samples of adult OCD patients compared to healthy controls and a peculiar gut microbiota composition in OCD patients (abundance of Rikenellaceae species and a relative decreased level of Prevotellaceae and two genera within the Lachnospiraceae) [24]. Interestingly, in this prospective study, the alpha diversity index of OCD patients became more similar to healthy controls after three months of pharmacological and cognitive behavioral treatment [24]. Moreover, in this study, the analysis of the difference in the microbiome composition of the oropharyngeal swab samples showed a difference in the Fusobacteria to Actinobacteria ratio between OCD patients and healthy controls [24]. The specific composition of the stool and oropharyngeal microbiota observed in this study further confirmed the abundance of species linked to inflammation (e.g., Rikenellaceae species have been linked to gut inflammation).

Consistently with these studies, a recent study analyzing the circulating bacterial extracellular vesicles in the serum of drug-free adults with OCD showed decreased alpha and beta diversity indices in OCD patients compared to healthy controls [25]. Interestingly, a sub-analysis on early-onset versus late-onset patients showed a significant difference in the genera Corynebacterium and Pelomonas in early-onset patients [25].

On the other hand, a recent study did not find a significant difference between OCD patients and controls on microbiota diversity index and species. Also, in this study, the 15 OCD patients who completed at least five sessions of exposure and response prevention therapy did not show any significant change in microbiota diversity and species after treatment [13]. Of note, compared to the other studies, in this study, half of the included patients were on stable medication regimes.

Interestingly, a recent study showed that the higher levels of oxytocin gene DNA methylation (inversely correlated with gene expression) observed in adult OCD patients compared to healthy controls is correlated to the higher presence of Actinobacteria in OCD saliva [26].

Finally, the presence of a pro-inflammatory gut microbiota composition (increase in Bacteroides, Odoribacter, and Oscillopora) has been shown in 4–8 yr old children with a diagnosis of PANS/PANDAS (pediatric acute-onset neuropsychiatric syndrome/neuropsychiatric disorders associated with streptococcal infection) [21].

Taken together, these data consistently point toward the presence of an altered gut microbiota composition, specifically related to a relative abundance of species with a known pro-inflammatory effect.

### 3.2. Preclinical Evidence of Gut Microbiota Alteration in OCD

The current literature on animal studies is quite limited, but it consistently reports an altered gut microbiota composition in animal models of OCD and an effect of gut microbiota manipulations on OCD-like phenotype expression.

The presence of altered gut microbiota composition in animal models of OCD was first described in a study on a natural model of OCD (the deer mouse model) [27]. Almost 30% of deer mice showed a naturally occurring OCD-like phenotype (large nest building). In this study, the authors showed a significant difference between the gut microbiota composition of OCD-like mice and control mice, with the latter showing a higher prevalence of anti-inflammatory species and the first showing a higher prevalence of pro-inflammatory species (Desulfovermiculus, Aestuariispira, Peptococcus, and Holdemanella) [27].

A recent relevant study suggested that gut microbiota transplantation from human OCD donors into mice can transfer OCD-relevant core behavioral features [28]. In this study, a sample of gut microbiota-depleted mice was transplanted with OCD patients’ or healthy controls’ gut microbiota through a fecal microbiota transplantation technique. Mice transplanted with OCD microbiota showed higher anxiety and compulsive-like behaviors compared to mice transplanted with control microbiota (e.g., reduced time spent in the open arms during the open field test and increased marble burying and digging and grooming behaviors). OCD-colonized mice showed gut dysbiosis with high levels of pro-inflammatory bacteria and a paucity of anti-inflammatory bacteria. Also, OCD-colonized mice showed some alterations of the medial prefrontal cortex (mPFC) compared to control mice (e.g., reduced c-fos expression, lower post-synaptic density and myelin thickness, and lower evoked neural firing). Interestingly, the authors found pro-inflammatory activation of the microglia in the OCD-colonized mice as well as inflammatory activation of the intestinal barrier, suggesting that neuronal anomalies resulting from gut dysbiosis are associated with neuroinflammation and result in abnormal behavior via gut–brain crosstalk. This gut–brain interaction seemed to be mediated by high levels of succinic acid. Indeed, OCD-colonized mice showed higher brain (mPFC) and serum levels of succinic acid, and this metabolite has been shown to reduce neural activity and display pro-inflammatory activities [28].

Concerning the effect of gut microbiota manipulation on OCD phenotypes, a recent study showed that the administration of a multistrain probiotic (Bifidobacterium lactis UBBLa-70, Bacillus coagulans Unique IS-2, Lactobacillus rhamnosus UBLR-58, Lactobacillus plantarum UBLP-40, Bifidobacterium infantis UBBI-01, Bifidobacterium breve UBBr-01, and glutamine) is able to abolish the OCD-like behaviors induced by quinpirole administration in rats [29]. Also, the probiotic formulation prevented the elevated mRNA expression of interleukin-6, tumor-necrosis factor-α, and C-reactive protein in the amygdala and dysregulated levels of 5-hydroxytryptamine, dopamine, and noradrenaline in the frontal cortex of quinpirole-injected rats. Moreover, the multistrain probiotic formula promoted colon integrity (the altered levels of goblet cells and cryptovilli ratio in quinpirole rats were prevented by multistrain probiotic treatment) [29].

The results of this recent study are in line with two previous studies showing the positive effects of probiotics administration on OCD-like behaviors in mice. Indeed, a study showed the efficacy of Lactobacillus casei administration in reversing OCD-like behaviors induced in quinpirole-administered rats [30], and a previous study showed the efficacy of Lactobacillus rhamnosus GG in preventing OCD-like behaviors induced by the administration of a 5HT1A/1B receptor agonist in rats [31].

### 3.3. Irritable Bowel Syndrome (IBS) and OCD

According to the current international diagnostic criteria (the Rome IV criteria), IBS is diagnosed on the basis of recurrent abdominal pain related to defecation or in association with a change in stool frequency or form, often accompanied by bloating. Symptoms must be chronic, occurring at least once per week, on average, in the previous 3 months, with a duration of at least 6 months [32]. United States prevalence varies across 7–16%, and recent meta-analyses showed a mean prevalence worldwide of 9.2% with the Rome III criteria and of 3.8% with the more restrictive Rome IV criteria [33]. IBS represents a prototype of a gut–brain axis disorder. Indeed, subjects with a genetic predisposition and an exposure to environmental stress factors (e.g., bowel infections, inflammation, exaggerated stress response associated with anxiety and depression, etc.) may develop an alteration of the intestinal permeability that could result in a cascade of events (e.g., infiltration of inflammatory cells, localized edema, and release of cytokines or chemokines) that lead the subject to develop IBS symptoms. On the one hand, the symptoms of IBS could have a negative psychological impact by exacerbating anxiety and depressive symptoms, while on the other hand, other pathophysiological factors, such as the release of inflammatory mediators and gut microbiota changes, could negatively impact several brain functions and further exacerbate psychiatric symptoms [32]. Moreover, the bidirectional gut–brain link in IBS is supported by epidemiological studies showing that in the general population, psychiatric symptoms precede gastrointestinal symptoms for half of the subjects, while they follow gastrointestinal symptoms in the other half [34]. Also, in recent years, several studies have suggested a genetic association between IBS and several psychiatric disorders (mood and anxiety disorder, psychoses, post-traumatic stress disorders, and ADHD) [35,36].

Up to date, only a few studies have investigated the prevalence and clinical correlates of IBS in OCD patients and vice versa. While an early study on a sample of OCD outpatients found a prevalence of 16% of IBS (with the Rome II criteria) that was considered comparable to that of the general population using the same criteria [37], two subsequent case–control studies found significantly higher prevalence in OCD patients compared to controls. Indeed, an early case–control study on a small sample of OCD patients (n: 37) found a prevalence of IBS (using the Rome I criteria) in OCD patients of 35.1% compared to 2.5% in healthy controls [38]. Also, a recent case–control study on 21 drug-free OCD patients and 22 controls showed that OCD patients have higher gastrointestinal symptoms than controls and found a prevalence of IBS (using the Rome III criteria) of 47.6% and 4.5 % in OCD patients and healthy controls, respectively (with diarrhea-predominant IBS as the most common subtype) [39]. Interestingly, in this study, OCD patients with comorbid IBS showed higher levels of inflammatory markers (IL-6) and higher levels of mood symptoms [39].

On the other hand, a larger study investigated the prevalence of psychiatric disorders in patients seeking treatment for IBS, and a recent study on a sample of 120 patients with IBS (according to the Rome II criteria) found a prevalence of comorbid IBS and OCD of 14.9%. Comorbid patients were more likely to be female, young (around their twenties), and single [40].

### 3.4. Inflammatory Bowel Disease (IBD) and OCD

Inflammatory bowel disease (IBD), containing Crohn’s disease (CD) and ulcerative colitis (UC), is a chronic systemic immune-mediated inflammatory disease that mainly affects the digestive system. The predominant clinical symptoms of IBD are abdominal pain and diarrhea, while a common symptom of UC is anal bleeding. In UC, inflammation is limited to colon mucosa and rectal mucosa, but in CD, inflammation involves any transmural segment of the entire digestive tract from mouth to anus [41]. IBD often occurs in early adulthood and is associated with subsequent colorectal cancer, stroke, and other diseases [42]. Psychiatric disorders, especially mood and anxiety disorders, are common comorbid conditions in IBD patients [43]. Interestingly, a recent relevant preclinical study shed light on the pathophysiological mechanisms linking IBD to anxiety and depressive symptoms by demonstrating a central role of the gut–brain axis in modifying the blood–brain barrier and consequently inducing psychiatric-like symptoms in animal models of IBD [44]. Thus, the relationship between OCD and IBD could be interesting when considering IBD also as a microbiome–gut–brain axis disorder.

The available studies on the prevalence of OCD in IBD patients showed a pooled prevalence of OCD of 9.4%, significantly higher than the 1–3% reported in the general population [45]. These results are in line with some recent clinical studies showing high rates of obsessive–compulsive symptoms in adult patients with IBD, especially in those with an active disease, and with an early report on children with IBD [46,47,48]. Although the nature of this association is still not clear, interestingly, in a recent study using a Mendelian randomization method, OCD and ulcerative colitis (UC) showed a significant causal bidirectional link [42]. Of note, a pilot study on 10 patients with IBD showed a decrease in OC symptoms after fecal microbiota transplantation [49].

## 4. Discussion

### 4.1. Current Evidence of Microbiome and Bowel Syndrome in OCD: Summary

Although the available literature on gut microbiota and gastrointestinal problems in OCD patients is still limited, it consistently points toward the presence of gut microbiome alterations and high rates of bowel syndromes in OCD patients. Indeed, both clinical and preclinical studies showed the presence of a gut microbiota alteration in OCD mainly characterized by a higher prevalence of pro-inflammatory species. Also, a recent preclinical study showed direct involvement of the human OCD gut microbiota composition in the induction of OC-like symptoms in animals [28]. On the other hand, clinical studies showed significantly high prevalence rates of OCD in patients affected by irritable bowel syndrome and inflammatory bowel disease, two prototypes of microbiome–gut–brain axis disorders [38,39,45,46,47,48].

### 4.2. Practical Implications for the Global Assessment of OCD

The emerging understanding of the interplay between gut health and psychiatric conditions, particularly OCD, suggests the potential role of the gut microbiota in influencing OCD symptomatology. In light of these insights, it is important to incorporate a comprehensive gastrointestinal assessment, encompassing factors that can impact the composition of the gut microbiota, into the “global assessment of OCD”. Such an assessment should address several aspects. Firstly, attention should be paid to gastrointestinal physical comorbidities and symptoms, including screening for symptoms indicative of irritable bowel syndrome (IBS) and inflammatory bowel disease (IBD), as well as consideration of baseline blood tests to evaluate markers of inflammation. Positive screening results may warrant further evaluation by a gastroenterologist.

Nutritional habits also merit exploration, particularly in individuals with OCD who may exhibit restricted eating patterns (or even have a confirmed diagnosis of ARFID) or “picky eating” behaviors, potentially influenced by OCD-related fears or rituals or comorbid conditions such as Autism Spectrum Disorders (ASDs). Additionally, assessing for reduced food intake due to the time spent during rituals and potential avoidance of certain food types, such as freshly cooked foods (due to contamination fears, which may lead to a preference for sealed or pre-prepared foods or microwaveable options), is also crucial. This is particularly important given the potential impact of dietary choices on the gut microbiome. It is possible that individuals with OCD may predominantly consume processed foods or added sugars that negatively influence gut health, either due to fear of contamination, spending excessive time on rituals that leave little time to prepare food, or avoidance of cutlery in cases where there are ego-dystonic obsessions about harming themselves or others, rather than adhering to a microbiome-friendly diet rich in fiber and fermented foods.

Furthermore, understanding bowel habits is essential, as individuals with OCD may intentionally reduce bowel frequency to avoid post-toilet rituals or experience diarrhea due to anxiety. Overuse of medications such as μ-receptor antagonists like loperamide to control bowel movements should also be explored, along with potential complications such as hemorrhoids or rectal prolapse. Additionally, assessing whether the patient is straining to achieve a “just right” sensation upon emptying the rectal ampulla is important, as this behavior may increase the risk of hemorrhoids and rectal prolapse, which has been reported in OCD [50]; these patients may tend to overuse laxatives, which will have an impact on the microbiome composition.

Fluid intake should be assessed, as patients with severe OCD may restrict fluid intake to avoid frequent toilet visits. Chronic fluid restriction in OCD not only poses risks to kidney health [18] but can also impact bowel function (i.e., lead to constipation) and the gut microbiome composition.

Exploring exercise habits is also important, considering that contamination fears may lead to avoidance of outdoor environments or gyms, impacting regular physical activity. Long periods of immobility can also be observed in OCD, especially in obsessive slowness. The need for lengthy shower rituals may also interfere with exercise routines, increasing the risk of constipation and, therefore, further affecting bowel habits.

Finally, in cases where there is a strong suspicion of a dysfunctional microbiota or gastrointestinal inflammation, consideration may be given to more detailed microbiome testing, providing insights into potential therapeutic targets.

### 4.3. Treatment Implications

The assessment of gastrointestinal problems and gut microbiota composition could have several potential treatment implications. For example, when facing an OCD patient with IBS or IBS-like symptoms, the presence of these symptoms could potentially guide the choice of the serotonergic agent. Indeed, several studies and recent meta-analyses showed that both SSRI and tricyclic antidepressants are efficacious for IBS symptoms despite the results of efficacy seeming more consistent for tricyclic antidepressants [51]. Thus, although clomipramine is usually not suggested as a first-line treatment, it could be considered as a first line in comorbid IBS and OCD patients, especially those with more prominent diarrhea, for whom the clomipramine’s anticholinergic effects could slow intestinal transit. On the other hand, for OCD-IBS patients with more predominantly constipation, SSRIs could be preferable because of their more pronounced effect of accelerating intestinal transit [51]. With regard to the treatment implications for microbiome assessment, the available evidence is still limited. However, in the last few years, some pilot trials showed promising results in the use of probiotics for the treatment of psychiatric conditions. For instance, recently, two different controlled trials on depressed patients using a multistrain probiotic as an add-on to antidepressant therapy showed superiority to placebo in both depressive and anxiety symptom improvement [52,53]. In the OCD field, there are still no studies directly assessing the effects of probiotics interventions. However, a few years ago, a study on healthy volunteers showed a beneficial effect on the obsessive–compulsive symptoms of a probiotic formula (Lactobacillus helveticus and Bifidobacterium longum) [54]. Interestingly, a meta-analysis showed that the probiotics administration of bifidobacterium species and Lactobacillus plantarum may be of benefit for IBS symptoms [55]. Thus, the use of probiotics (e.g., multistrain probiotics formula containing bifidobacterium and lactobacillus species) could be a reasonable add-on in OCD patients with comorbid gastrointestinal problems and IBS-like symptoms.

Lifestyle and dietary interventions should also be considered in the treatment plan of IBS-OCD patients. The consistent literature on IBS suggests adopting a diet rich in soluble fiber (psyllium husk) and a low-FODMAPs diet (low fermentable oligosaccharides, disaccharides, monosaccharides, and polyols, which are present in stone fruits, legumes, lactose-containing foods, and artificial sweeteners) [32]. Also, although with a more limited amount of evidence, physical exercise has been shown to improve IBS and OCD symptoms [56,57].

### 4.4. Limitation of the Current Literature

The interpretation of the current literature should be considered in light of several limitations. Despite the presence of several preclinical studies, clinical studies on gut microbiota in OCD patients are still limited and characterized by small samples. Also, there is a lack of longitudinal studies that could elucidate the effects of serotonergic medications on gut microbiota alterations. Finally, the studies focusing on inflammatory bowel diseases and irritable bowel syndrome are limited. The putative pathophysiological link between these disorders remains speculative, and again, there is a lack of longitudinal studies investigating the trajectory of these disorders in patients undergoing proper treatment for OCD.

## 5. Conclusions

In summary, while the current literature on gut microbiota and gastrointestinal issues in patients with OCD remains limited, emerging evidence suggests gut microbiome alterations and high rates of bowel syndromes in this population. These findings highlight the potential influence of the gut microbiota on OCD symptomatology and emphasize the importance of incorporating comprehensive gastrointestinal assessments into the “global assessment of OCD”. Such assessment should encompass various factors, including gastrointestinal physical comorbidities and symptoms, nutritional habits, bowel habits, fluid intake, exercise patterns, and potential microbiome dysfunctions or inflammation. Considering the treatment implications, interventions targeting gut health, such as probiotics and dietary modifications, may hold promise in improving symptoms in OCD patients with comorbid gastrointestinal problems. Further research in this area is warranted to better understand the interplay between gut health and OCD and to explore the effectiveness of targeted interventions in improving clinical outcomes.

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
