# Peer review of "Gut Microbiota and Gastrointestinal Symptoms in the Global Assessment of Obsessive–Compulsive Disorder: A Narrative Review of Current Evidence and Practical Implications"

_brainsci, 2024, doi:10.3390/brainsci14060539_

Round 1
Reviewer 1 Report
Comments and Suggestions for Authors
Brief summary: This interesting article provide a narrative review on the potential role of the gut microbiom and the clinical presentation of obsessive-compulsive disorder. The authors report that there is still a limited body of evidence on this topic and that gut microbiome alterations (and high rates of bowel syndromes) are identified in the population of patients with OCD. They support that these findings can be beneficial for treatment implications via the means of interventions that target gut health (probiotics, dietary modifications) as they could improve OCD symptoms. Please see my comments below.
Title: please add the notion of narrative review to the title to better situate the readership.
Abstract: It would be interesting to add one line as to how data was collected (how was the ltierature review conducted).
Introduction:
- While the first paragraph is beautifully written, it would be beneficial for the readership to have a short definitions of the main symptomns found in OCD (obsessions and compulsions) (line 29-43) as there exists a body of literature mapping these to gut microbiome implications.
- Please expand on lines 46-48 as ''novel mechanisms'' is unclear as well as ''new avenues''.
- While lines 56-62 suggests the importance of conducting this review, it is unclear to the readership why it is important to understand better these implications for patients suffering from OCD. This should be better highlighted.
- Line 67 is also true for Anorexia Nervosa and Bulimia which are much more impairing than ARFID.
- It would be interesting if the authors add their own hypotheses at the end of the last paragraph (lines 76-85) considering their expertise.
Materials and Methods:
- Why was a narrative review conducted rather than a scoping review considering the number of studies in the field?
- Inclusion criteria and exclusion criteria should be better defined.
- Please include the search strategy used.
- How was data collected (Excel? Endnote?)
- It is unclear how data was analysed and what was analysed.
- Overall this section should be expanded to reflect the steps taken to conduct the search and the analysis.
Results:
- No comment. The results are very well presented in my opinion.
Discussion:
- Please include the limitations of this study.
- Discussion is also very well presented.
Comments on the Quality of English LanguageNil
Author Response
Reviewer #1:
Reviewer comment: Title: please add the notion of narrative review to the title to better situate the readership.
Answer to reviewer: we thank the reviewer for this comment. We modified the title as suggested: “Gut microbiota and gastrointestinal symptoms in the global assessment of obsessive-compulsive disorder: a narrative review of current evidence and practical implications. “
Reviewer comment: Abstract: It would be interesting to add one line as to how data was collected (how was the literature review conducted).
Answer to reviewer: we add a line as suggested : “ For this purpose, a search on studies focusing on OCD, gut microbiota, irritable bowel syndrome and inflammatory bowel disease was conducted by two independent reviewers. “
Reviewer comment: Introduction: - While the first paragraph is beautifully written, it would be beneficial for the readership to have a short definitions of the main symptomns found in OCD (obsessions and compulsions) (line 29-43) as there exists a body of literature mapping these to gut microbiome implications.
Answer to reviewer: many thanks for pointing this out. The main symptoms of OCD have now been added: “OCD is characterized by obsessions and compulsions. Obsessions are intrusive, repetitive, and distressing thoughts, images, or urges that are often perceived as unrealistic or excessive by the individual experiencing them. Compulsions are repetitive and often ritualized behaviors or mental acts that individuals feel compelled to perform in response to an obsession or according to rigidly applied rules. The primary purpose of compulsions is to alleviate the anxiety or distress triggered by obsessions.These lead to significant distress or impairment, are time-consuming, and the individual often tries to resist or control them. “
Reviewer comment: - Please expand on lines 46-48 as ''novel mechanisms'' is unclear as well as ''new avenues''.
Answer to reviewer: Many thanks for your very helpful observation. The text has been changed to: This emerging field holds promise for exploring new pathways through which the gut microbiota affect brain function, such as modulating neurotransmitter levels and influencing inflammatory and immune responses. These interactions may subsequently influence the development and maintenance of psychiatric conditions. Understanding these mechanisms could provide insights into developing alternative therapeutic strategies targeting the microbiome, including the use of probiotics, prebiotics, fecal transplants, and dietary interventions.
Reviewer comment: - While lines 56-62 suggests the importance of conducting this review, it is unclear to the readership why it is important to understand better these implications for patients suffering from OCD. This should be better highlighted.
Answer to reviewer: many thanks for your comment which we believe we have addressed and we hope will provide more rationale as to why this is particularly important on OCD : These observations raise intriguing questions about the potential influence of psychiatric symptomatology on gut microbial composition, and vice versa. Although the studies cited above, along with several others, have demonstrated an involvement of the gut microbiome in various psychiatric disorders, its specific role in OCD remains not fully understood.
The authors chose to focus on the role of the gut microbiome in OCD for several key reasons. Understanding this relationship could be crucial for patients suffering from OCD, as a substantial proportion fail to respond to existing treatments. Investigating the role of the gut microbiome in OCD could lead to more effective and comprehensive treatment strategies, potentially benefiting those patients who have not responded to treatment or who continue to experience significant residual symptoms. Additionally, the symptomatology and phenomenology of OCD, including its comorbidities, provide strong grounds for such an investigation. OCD symptoms may contribute to the onset and maintenance of alterations in the gut microbiome through various mechanisms. For instance, compulsions such as excessive washing and cleaning can reduce exposure to diverse microbial species encountered in the environment on a daily basis [13]. Furthermore, OCD is often associated with altered eating habits, which can influence microbiome composition. Eating disorders show comorbidity rates with OCD of 17% (Piggott et al., 1994), with a lifetime comorbidity of 19% in Anorexia Nervosa and 14% in Bulimia Nervosa (Mandelli et al., 2020). Selective and restricted food intake (picky eating) has also been associated with OCD [14]. Conversely, individuals exhibiting picky eating behaviors that meet the diagnostic criteria for Avoidant/Restrictive Food Intake Disorder (ARFID) tend to exhibit a higher prevalence of OCD symptoms [15].
Additionally, patients with OCD may engage in elaborate and time-consuming compulsions following bowel movements, often resulting in intentionally decreasing the frequency of bowel movements. This reduction may be accompanied by food restriction, or alterations in diet, serving as avoidance strategies. For instance, Chen et al. [13] demonstrated reduced fiber intake among patients with OCD. Similarly, patients with OCD may restrict fluid intake, which, besides increasing the risk of chronic kidney failure [16], can cause constipation and subsequently impact the composition of the gut microbiome.
Reviewer comment: - Line 67 is also true for Anorexia Nervosa and Bulimia which are much more impairing than ARFID.
Answer to reviewer:Many thanks for highlighting this omission. The text has bee modified as follows: Furthermore, OCD is often associated with altered eating habits, which can influence microbiome composition. Eating disorders show comorbidity rates with OCD of 17% (Piggott et al., 1994), with a lifetime comorbidity of 19% in Anorexia Nervosa and 14% in Bulimia Nervosa (Mandelli et al., 2020). Selective and restricted food intake (picky eating) has also been associated with OCD [14]. Conversely, individuals exhibiting picky eating behaviors that meet the diagnostic criteria for Avoidant/Restrictive Food Intake Disorder (ARFID) tend to exhibit a higher prevalence of OCD symptoms [15].
Reviewer comment: - It would be interesting if the authors add their own hypotheses at the end of the last paragraph (lines 76-85) considering their expertise.
Answer to reviewer: many thanks for giving us this opportunity for further reflection. We now added the following paragraph after line 85 : We hypothesize that OCD symptomatology, such as excessive cleaning and food and fluid restriction, may contribute to alterations in the gut microbiome, and that specific changes in gut microbiota composition may, in turn, contribute to OCD. These changes could potentially serve as biomarkers for targeted interventions. This would render the assessment of eating habits and gastrointestinal symptoms an essential component of the "global assessment" of OCD. Additionally, we propose that interventions aimed at modulating the gut microbiome, such as dietary adjustments, probiotics, and prebiotics, may alleviate certain OCD symptoms and improve overall treatment outcomes. Further research in this area could pave the way for novel, microbiome-targeted therapies that complement existing OCD treatments.
Reviewer comment: Materials and Methods: - Why was a narrative review conducted rather than a scoping review considering the number of studies in the field?
Answer to reviewer: We thank the reviewer for this comment and we agree that a more systematic revision of the literature could be useful. The initial purpose of the study was just to summarize and present the available research on the topic of OCD, microbiome and intestinal problems in order to stress the clinical relevance of this emerging topic. Reviewer comment: - Inclusion criteria and exclusion criteria should be better defined.
Answer to reviewer: we added the inclusion and exclusion criteria to the methods section as suggested “Inclusion criteria were original research articles on OCD and microbiome, OCD and irritable bowel syndrome or OCD and inflammatory bowel disease. We included all articles on these topics, regardless of whether they were animal studies, genetic studies or studies on clinical populations.
Exclusion criteria were non-original articles (e.g., reviews) and articles that did not examine the link between OCD, microbiome, irritable bowel syndrome and inflammatory bowel disease. Also, articles in a language other than English and published in non peer-review journals were excluded. “
Reviewer comment: - Please include the search strategy used. How was data collected (Excel? Endnote?)
Answer to reviewer: we add the search strategy as suggested: Screening process by application of those criteria to title/abstracts of the articles was carried out by the two authors (GG and IP). An agreement could be achieved concerning the few discrepancies in title/abstract screening. This resulted in 28 articles being selected and inserted in a shared excel sheet for full text screening which reduced the final number of eligible articles to 19.”
Reviewer comment: - It is unclear how data was analysed and what was analysed.
Answer to reviewer: we further clarify how the collected studies were grouped and then analyzed: “It was further decided to group the articles by methods employed in these studies as they are extremely heterogeneous content-wise . This decision led to the following sections: clinical studies of gut microbiota, preclinical studies of gut microbiota, clinical studies on irritable bowel syndrome and OCD, clinical studies on inflammatory bowel disease and OCD.”
Reviewer comment: - Overall this section should be expanded to reflect the steps taken to conduct the search and the analysis.
Answer to reviewer: we expanded this section as suggested
Reviewer comment: Discussion: - Please include the limitations of this study.
Answer to reviewer: we included a new section as suggested:
4.4 Limitation of the current literature
The interpretation of the current literature should be considered in light of several limitations. Despite the presence of several preclinical studies, clinical studies on gut microbiota in OCD patients are still limited and characterized by small samples. Also, there is a lack of longitudinal studies that could elucidate the effects of serotonergic medications on gut microbiota alterations. Finally, the studies focusing on inflammatory bowel diseases and irritable bowel syndrome are limited. The putative pathophysiological link between these disorders remains speculative and again there is a lack of longitudinal studies investigating the trajectory of these disorders in patients undergoing a proper treatment for OCD.

Reviewer 2 Report
Comments and Suggestions for Authors
Thanks for the useful article.
One of the strengths of this article is attention to the accompanying physical and mental disorders, especially digestive diseases, which are also considered in genetic studies.
-Genetic articles on the relationship between OCD and GI diseases should also be studied.
- The cited articles are mostly related to animal studies. If possible, more articles related to human studies should be added.
-A more complete explanation should be provided regarding the reason for choosing obsessive-compulsive disorder among other psychiatric disorders.
Author Response
Reviewer#2
Reviewer comment:-Genetic articles on the relationship between OCD and GI diseases should also be studied.
Answer to reviewer: We thank the reviewer for this relevant comment. We added a sentence on the studies on the genetic overlap between psychiatric disorders and IBS as suggested: “Also, in recent years, several studies suggested a genetic association between IBS and several psychiatric disorders (mood and anxiety disorder, psychoses, post-traumatic stress disorders, ADHD) [56-57].”
Reviewer comment:- The cited articles are mostly related to animal studies. If possible, more articles related to human studies should be added.
Answer to reviewer: We performed a new review process and we add a new study on OCD patients in the clinical evidence of gut microbiota alteration in OCD section: …”Consistently with these studies, a recent study analyzing the circulating bacterial extracellular vesicles in the serum of drug-free adults with OCD showed decreased alpha and beta diversity indices in OCD compared to healthy controls [52]. Interestingly, a sub analysis on early-onset versus late-onset patients showed a signficant difference on the genera Corynebacterium and Pelomonas in early-onset patients [52].” We also added a study on microbiota transplantation in inflammatory bowel disease: Of note, a pilot study on 10 patients with IBD, showed a decrease of OC symptoms after fecal microbiota transplantation [53].
Reviewer comment:-A more complete explanation should be provided regarding the reason for choosing obsessive-compulsive disorder among other psychiatric disorders.
Answer to reviewer: many thanks for this comment; we believe we addressed this point by changing the introduction as follows: “ These observations raise intriguing questions about the potential influence of psychiatric symptomatology on gut microbial composition, and vice versa. Although the studies cited above, along with several others, have demonstrated an involvement of the gut microbiome in various psychiatric disorders, its specific role in OCD remains not fully understood. The authors chose to focus on the role of the gut microbiome in OCD for several key reasons. Understanding this relationship could be crucial for patients suffering from OCD, as a substantial proportion fail to respond to existing treatments. Investigating the role of the gut microbiome in OCD could lead to more effective and comprehensive treatment strategies, potentially benefiting those patients who have not responded to treatment or who continue to experience significant residual symptoms. Additionally, the symptomatology and phenomenology of OCD, including its comorbidities, provide strong grounds for such an investigation. OCD symptoms may contribute to the onset and maintenance of alterations in the gut microbiome through various mechanisms. For instance, compulsions such as excessive washing and cleaning can reduce exposure to diverse microbial species encountered in the environment on a daily basis [13]. Furthermore, OCD is often associated with altered eating habits, which can influence microbiome composition. Eating disorders show comorbidity rates with OCD of 17% (Piggott et al., 1994), with a lifetime comorbidity of 19% in Anorexia Nervosa and 14% in Bulimia Nervosa (Mandelli et al., 2020). Selective and restricted food intake (picky eating) has also been associated with OCD [14]. Conversely, individuals exhibiting picky eating behaviors that meet the diagnostic criteria for Avoidant/Restrictive Food Intake Disorder (ARFID) tend to exhibit a higher prevalence of OCD symptoms [15].
Additionally, patients with OCD may engage in elaborate and time-consuming compulsions following bowel movements, often resulting in intentionally decreasing the frequency of bowel movements. This reduction may be accompanied by food restriction, or alterations in diet, serving as avoidance strategies. For instance, Chen et al. [13] demonstrated reduced fiber intake among patients with OCD. Similarly, patients with OCD may restrict fluid intake, which, besides increasing the risk of chronic kidney failure [16], can cause constipation and subsequently impact the composition of the gut microbiome.”
